# Evaluation of Patient Benefits from the Superficial Circumflex Iliac Artery Perforator Flap in Elderly Patients

**DOI:** 10.3390/bioengineering12040394

**Published:** 2025-04-07

**Authors:** Hongmin Luo, Huining Bian, Zuan Liu, Chuanwei Sun, Hanhua Li, Lianghua Ma, Xiaoyan Wang, Zhifeng Huang, Xu Mu, Shenghua Chen, Yuyang Han, Lin Zhang, Shaoyi Zheng, Zeyang Yao, Wen Lai

**Affiliations:** 1Department of Burns and Wound Repair Surgery, Guangdong Provincial People’s Hospital (Guangdong Academy of Medical Sciences), Southern Medical University, Guangzhou 510080, China; luohongmin@gdph.org.cn (H.L.); bianhuining@hotmail.com (H.B.); wylandlza@163.com (Z.L.); chuanwei.sun@163.com (C.S.); judeyylhh@126.com (H.L.); 13580551814@163.com (L.M.); wangxy48@mail2.sysu.edu.cn (X.W.); 13802938068@163.com (Z.H.); mx3212348451@163.com (X.M.); 17387204193@163.com (S.C.); 13711807531@163.com (Y.H.); m13833863697_1@163.com (L.Z.); zsysskbox@163.com (S.Z.); 2School of Medicine, South China University of Technology, Guangzhou 510040, China

**Keywords:** superficial circumflex iliac artery perforator flap, reconstructive surgery, elderly patients, flap survival, aging population, elder patients

## Abstract

Background: The superficial circumflex iliac artery perforator (SCIP) flap is widely recognized for its reliability and minimal donor site morbidity in reconstructive surgery. However, its safety and efficacy in elderly patients—a growing demographic with increased comorbidities—remain less understood. This study aims to evaluate the clinical outcomes of the SCIP flap in elderly patients compared to younger patients, focusing on flap survival, complications, and recovery. Methods: In this retrospective cohort study, conducted at Guangdong Provincial People’s Hospital, from 28 August 2019 to 7 June 2024, we included 37 patients who underwent SCIP flap procedures for reconstruction. Patients were divided into two groups: younger (15–59 years) and elderly (≥60 years). Key variables analyzed included demographics, comorbidities, flap characteristics, recipient sites, arterial sources, and surgical outcomes. Univariate analysis and ROC curve analysis were used to explore the impact of age on flap survival and complications. Results: The cohort consisted of 28 younger and 9 elderly patients. Vascular disease was significantly more prevalent in the elderly group (88.9% vs. 21.4%, *p* = 0.001), and abnormalities in the CTA results indicate that the elderly cohort exhibited a 29-fold increased odds of vascular disease compared to younger patients (OR = 29.17, 95% CI: 4.82–176.40, *p* = 0.001). However, no significant differences were found between the groups in terms of flap area, recipient sites, or arterial sources. Hospital stay duration and flap survival rates were comparable across both age groups, with no cases of total flap loss reported. While systemic complications were somewhat higher in the elderly group, this difference did not reach statistical significance. The ROC analysis (AUC = 0.52) indicates that age alone is not a significant predictor of flap survival. Conclusions: The SCIP flap is a safe and effective reconstructive option for elderly patients, despite a higher incidence of vascular disease. Flap survival and postoperative recovery were favorable, indicating that the procedure is viable for older patients. These findings support the continued use of SCIP flaps in aging populations, emphasizing the need for individualized surgical approaches to optimize patient outcomes.

## 1. Introduction

From a global healthcare perspective, the continuous advancement in microsurgical techniques and improvements in microsurgical equipment have ushered in an era of supermicrosurgery [1]. Free tissue transfer has become a cornerstone of the reconstruction of skin and soft-tissue defects caused by conditions such as tumors, trauma, and infections. Among these techniques, the superficial circumflex iliac artery perforator (SCIP) flap has gained significant recognition and popularity in recent years, emerging as a key method in reconstructive surgery and a workhorse flap [2,3] for the reconstruction of skin and soft-tissue defects resulting from various causes. The increasing reliance on SCIP flaps reflects a broader trend in global healthcare toward adopting innovative, minimally invasive surgical options that optimize patient outcomes.

The SCIP flap was first described by Koshima, in 2004 [4], and has since become a versatile option for reconstructive surgery, particularly for complex defects requiring reliable tissue coverage with minimal donor site morbidity. This flap has gained attention for its utility in reconstructive procedures because of its reliable and predictable vascular anatomy, as well as its relatively straightforward harvesting compared to other free flaps. In recent years, experts have continually proposed new techniques and strategies for harvesting SCIP flaps [5,6,7,8,9,10,11,12]. These advancements have made the harvesting process increasingly simple and safe, and it is now considered one of the “best new flaps” [13]. However, despite its many advantages, the SCIP flap has some drawbacks, including small-caliber vessels, a short vascular pedicle, and variability in the location of perforating vessels [2]. Consequently, its use in elderly patients—a rapidly growing demographic worldwide—raises concerns about its safety and efficacy. These concerns stem from both the limitations of the flap itself and the increased prevalence of comorbidities and age-related physiological changes.

With the global aging population on the rise, especially in developed countries, the proportion of elderly individuals is increasing substantially. For example, in the United States, individuals aged 65 and older currently make up about 17% of the population, a figure expected to rise significantly in the coming decades. This demographic shift means that an increasing number of reconstructive procedures, including free tissue transfers like the SCIP flap, will be performed on older patients.

Despite the advancements in surgical techniques and perioperative care, elderly patients often present unique challenges, including reduced physiological reserve and increased risk of postoperative complications. The SCIP flap, though generally well-tolerated, must be scrutinized for its safety and effectiveness in this patient group. Previous studies have shown that while age itself is not always a direct predictor of poor outcomes in free flap transfer, the presence of comorbid conditions and overall health status play crucial roles in surgical success [14]. Another concern with using the SCIP flap in elderly patients is the risk of anastomotic thrombosis due to the small caliber of the superficial circumflex iliac artery (SCIA) and the presence of vascular sclerosis, which is more common among this age group.

The safety and efficacy of microsurgery for elderly patients have undergone a paradigm shift in recent decades. Large-scale studies, such as a breast reconstruction, including 445 free flaps (≥65 years), and head/neck reconstruction series, demonstrate that age alone does not compromise flap survival or functional outcomes when modern protocols are applied [15,16,17,18,19]. Despite robust evidence supporting the safety and efficacy of microsurgical reconstruction in elderly patients across some anatomical contexts, the safety and efficacy of SCIP as applied to elderly patients are still under-researched. Thus, this study aims to evaluate the safety and efficacy of the SCIP flap specifically in elderly patients through a single-center retrospective analysis. We seek to understand the outcomes of SCIP flap procedures in this age group, assess the impact of age-related factors on flap survival and postoperative recovery, and identify key predictors of complications. By doing so, we hope to provide insights into optimizing surgical strategies and improving patient care for this growing and vulnerable population.

## 2. Materials and Methods

### 2.1. Study Design

This retrospective study was conducted at Guangdong Peoples Hospital and reviewed all patients who underwent SCIP flap procedures for reconstruction between 28 August 2019 and 7 June 2024. The study focused on evaluating the safety and efficacy of SCIP flaps in elderly patients compared to younger patients.

### 2.2. Patient Selection

In this study, “older patients” are defined as being ≥60 years old. Therefore, patients were divided into the following two groups: Young group, aged 15–59 years, and the Elder group, ≥60 years old, based on the age at the time of surgery. The inclusion criteria included patients who underwent SCIP flap procedures within the specified period. Patients with incomplete medical records or those undergoing other types of free flap procedures were excluded.

### 2.3. Data Collection

Data were collected from medical records and included the following:Demographic information: age, sex, etc.;Preoperative variables: body mass index (BMI), comorbidities (e.g., diabetes and hypertension), preoperative treatments (e.g., radiation and chemotherapy), flap type, recipient site, surgery duration, hospital stay, and, especially, preoperative CTA information was collected to evaluate recipient vessel quality (e.g., patency and calcification) in patients with suspected vascular disease, while handheld Doppler ultrasonography was used for SCIP perforator localization;Outcomes: flap survival and postoperative complications.

### 2.4. Statistical Analysis

Data were analyzed using R software (version [4.1.0], R Core Team, Vienna, Austria). Descriptive statistics were calculated for both age groups, including means, standard deviations, and frequencies. The following were used for the comparative analysis between the Young and Elder groups: univariate analysis—differences in flap survival and postoperative complications between the two groups were assessed using Fisher’s exact test for categorical variables and *t*-tests or Mann–Whitney U tests for continuous variables, as appropriate. Logistic regression models were used to assess the impact of age on flap survival and postoperative complications. Flap survival and complication were the primary endpoints. Statistical significance was set at *p* < 0.05. The Akaike information criterion (AIC) was used for model the selection to ensure the best fit. For the ROC curve analysis, we evaluated the ability of age to predict flap reconstruction outcomes, specifically the complete survival of the flap. The area under the curve (AUC) was calculated to assess the predictive performance.

### 2.5. Ethical Considerations

The study has been approved by the Ethics Committee of Guangdong Provincial People’s Hospital (ethics number: KY-Z-2021-613-03), and all procedures adhered to ethical guidelines for research involving human subjects, as per Guangdong People’s Hospital policies. No direct patient consent was required due to the nature of the study and its reliance on de-identified patient data.

## 3. Results

Finally, a total of 37 patients were selected for the study cohort. Table 1 presents the baseline characteristics of the cohort, which includes 37 individuals divided into the following two groups: Young (15–59 years) and Elder (≥60 years). The analysis shows that the Young group had a significantly lower mean age compared to the Elder group (41.07 years vs. 63.78 years, *p* < 0.001). There were no significant differences between the two groups in terms of gender distribution, height, weight, and BMI. Specifically, the proportions of males, as well as the height, weight, and BMI values, were similar across both groups (*p* > 0.05 for all).

Our study also summarized the prevalence of concomitant diseases in the patient cohort in Table 2, with the data split between the Young (15–59 years) and Elder (≥60 years) groups. The incidences of tumors, diabetes mellitus, and infections were comparable between the two groups, with no significant differences. However, there was a marked difference in the prevalence of vascular disease, which was significantly higher in the Elder group (88.9%) compared to the Young group (21.4%), with a *p*-value of 0.001. Additionally, lung disease was present in 11.1% of Elder patients but not in Young patients, though this difference was not statistically significant. The prevalence of other concomitant diseases was higher in the Young group but did not show a significant difference from the Elder group.

The mean flap area for the overall cohort was 80.10 cm^2^ (SD = 55.51), with no significant difference between Young patients (84.50 cm^2^, SD = 60.58) and Elder patients (66.92 cm^2^, SD = 35.98) (*p* = 0.418). Regarding recipient sites, there were no significant differences between the two groups. The distribution of recipient sites was as follows: Head/neck/trunk (24.3% overall), lower extremities (59.5% overall), and upper extremities (16.2% overall). The Elder group had a slightly higher percentage of lower extremity reconstructions compared to the Young group, though this difference was not statistically significant (*p* = 0.297).

The reasons for surgery presented in Table 3 show no significant differences (*p* = 0.247), with infections and complications being the most common indications. The CTA results reveal a notable disparity: 77.8% of elder patients had abnormalities compared to only 10.7% in young patients. This indicates the elderly cohort exhibited 29-fold increased odds of vascular disease compared to younger patients (OR = 29.17, 95% CI: 4.82–176.40, *p* = 0.001), corroborating age as a dominant driver of other vascular diseases such as atherosclerosis. Nevertheless, this did not result in inferior flap outcomes, likely due to protocolized CTA evaluation, which facilitated the selective use of non-calcified vessels. These findings underscore that vascular disease prevalence, while epidemiologically significant, becomes surgically surmountable through meticulous planning—a critical distinction when counseling elderly patients.

There were no significant differences in recipient artery selection between the two groups. The most commonly used recipient arteries were the posterior tibial artery (16.2% overall), superficial temporal artery (16.2% overall), and dorsalis pedis artery (24.3% overall). The recipient artery selection was similar between the Young and Elder groups (*p* = 0.445). Regarding anastomosis methods, end-to-end arterial anastomosis was performed in 59.5% of cases overall, with a non-significant trend toward more frequent use in Elder patients (77.8% vs. 53.6% in Young patients, *p* = 0.37). The median number of venous anastomoses was 2.00 (IQR 1.00–2.00) for the overall cohort, with no significant difference between the groups (*p* = 0.181). While there were some differences in terms of recipient sites and recipient arteries, these were not statistically significant. Detailed results of flap area, recipient sites, and recipient arteries used in the reconstruction procedures, segmented by age groups, are presented in Table 4.

For surgery outcomes, Table 5 shows that the average length of hospital stay was similar between Young and Elder patients. Flap survival rates were high and comparable across both groups, with no cases of total flap loss reported. Flap exploration procedures were performed more frequently in Young patients, but this difference was not statistically significant. Donor site complications were rare and did not differ between groups, while systemic complications were slightly more common in the Elder group, though not significantly. There were no patient deaths in either group.

Our results also show there were no significant differences between the two age groups in terms of flap survival, flap exploration, or complications. The univariate analysis indicated similar outcomes for complete flap survival and flap exploration, with no significant age-related effects. Donor site complications were rare and not interpretable due to their low occurrence in the Elder group. Systemic complications were slightly more frequent in the Elder group but did not reach statistical significance. Overall, these results suggest that the procedure’s safety and effectiveness are consistent across age groups.

The ROC curve analysis for predicting flap reconstruction outcomes based on age is showed on Figure 1, and an AUC almost equivalent to random chance (AUC = 0.52) was detected, indicating that age alone is not a significant predictor of flap survival. This suggests that age does not strongly influence the likelihood of complete flap survival in our study.

## 4. Discussion

The safety and efficacy of microsurgery in elderly patients have undergone a paradigm shift in recent decades. Early studies [20] reported alarmingly high mortality (4.4%) and surgical complication rates in cohorts arbitrarily defined as “elderly” (age > 50 years), largely attributed to unrefined surgical techniques, inadequate patient selection, and limited perioperative optimization. In stark contrast, modern studies [21,22] and our findings demonstrate that chronological age alone no longer predicts surgical outcomes when multidisciplinary advancements are leveraged. Specifically, our SCIP flap cohort achieved comparable flap survival (88.9% elderly vs. 85.7% younger, *p* = 0.999) and reoperation rates (11.1% vs. 25.0%, *p* = 0.678), with zero mortality—aligning with contemporary data showing <1% mortality in optimized elderly patients. These improvements in mortality and complication rates, along with the increased safety of microsurgery, can be attributed to advancements in patient optimization, surgical techniques, reduced surgical duration, and improved anesthesia management.

Despite robust evidence supporting the safety and efficacy of microsurgical reconstruction in elderly patients across diverse anatomical contexts, medical complications persist as a critical residual challenge, particularly in populations with advanced comorbidities. Large-scale studies such as breast reconstruction with 445 free flaps (≥65 years) [15] and head/neck reconstruction series [16,17,18,19] demonstrate that age alone does not compromise flap survival or functional outcomes when modern protocols are applied. Our findings extend this paradigm to the SCIP flap—a technically nuanced yet physiologically considerate option for elderly patients. While prior lower limb studies [23] validated free flap safety in extremity defects, the SCIP flap’s unique advantages (e.g., minimal donor morbidity, and adaptability to calcified recipient vessels via CTA-guided planning) remain underexplored in geriatric cohorts. In our series, elderly patients achieved comparable flap survival (88.9% vs. 85.7%, *p* = 0.999) and hospitalization durations (19.3 vs. 21.2 days, *p* = 0.419) to younger counterparts despite higher vascular comorbidities (88.9% with CTA-identified pathology), aligning with broader trends of optimized microsurgical safety [24]. These results position the SCIP flap as a geriatric-tailored solution, particularly for limb salvage where thin pliable tissue and reduced donor morbidity are paramount.

Thus, microsurgical feasibility in elderly patients needs to be well-established. In this study we evaluated the safety and effectiveness of the SCIP flap in elderly patients by comparing various factors between the Young (15–59 years) and Elder (≥60 years) groups. The study is enhanced by the inclusion of not only patients undergoing head and neck reconstruction but also those receiving upper and lower limb reconstruction. Our study provides compelling evidence that free SCIP flap can be performed safely and reliably in elderly patients.

In this study, the first question that needs to be addressed is how to define elderly individuals. Currently, various countries possess distinct criteria for defining the elderly. In developed countries, being or over 65 years old or even 70 years old is regarded as elderly. Most studies have set the cut-off age for the elderly at 60 to 70 years [25,26,27,28]. Other authors focus on patients >80 or 90 years old [29,30,31]. However, China remains a developing country. Our economic status, public health standards, and medical care still exhibit certain disparities when compared to developed nations. In China, individuals aged 60 and above are classified as elderly. Consequently, in this study, elderly patients are defined as those who are ≥60 years old. While the recent literature has advocated for finer age stratification (e.g., 65–74 vs. ≥75 years), our cohort’s demographic skew toward younger elderly patients precludes such subgroup analyses. Multicenter studies targeting octogenarians are urgently needed to validate our findings in late-elderly populations.

The demographic and clinical characteristics in Table 1 show that both age groups were similar in terms of gender distribution, height, weight, and BMI, with only age significantly differing between the groups. This similarity in baseline characteristics reinforces the comparability of the groups, supporting the validity of our comparisons regarding flap outcomes and complications.

One concern regarding the use of free flaps in elderly patients is their ability to tolerate prolonged anesthesia due to diminished cardiorespiratory function. The SCIP flap can be harvested from the superficial fascia layer without the need for muscle dissection. Moreover, various techniques for SCIP flap harvesting have been proposed by experts, making the procedure increasingly straightforward [8,9,10,11,12]. With adequate training, it can generally be completed within one hour. Therefore, a major advantage of the SCIP flap is its short harvesting time, which significantly reduces the impact of prolonged anesthesia on elderly patients. As outlined in Table 2, the prevalence of concomitant diseases varied notably between age groups. Elderly patients had a significantly higher incidence of vascular disease compared to younger patients, which aligns with known age-related health conditions. While there was no difference in prolonged hospitalization of elderly patients, despite their higher vascular disease burden, this may reflect our institutional protocols prioritizing geriatric perioperative care (e.g., early mobilization and multidisciplinary pain management). However, the single case of postoperative depression in the elderly group underscores the need for holistic psychological support in this population. Larger cohorts are required to determine whether age independently predicts psychiatric complications.

Despite this, systemic complications, though slightly higher in the Elder group, were not significantly different, suggesting that elderly patients can undergo this procedure with a manageable risk profile. The length of hospital stay was comparable between Young and Elder patients, indicating that age does not adversely affect overall recovery associated with this procedure. Although there is no specific literature assessing the safety and efficacy of SCIP flap in elderly patients, some studies have examined the utilization of free flaps in this population. Our results are consistent with finding from other studies on the use of various free flaps in elderly patients [14,32,33].

Various factors such as flap sizes, recipient site locations, and the selection of recipient vessels can potentially influence the outcome of the flap. Therefore, we conducted an analysis of these factors as well. Table 4 demonstrates that the flap area, recipient sites, and recipient vessels used were comparable across age groups. While flap areas and types of recipient vessels varied, there were no significant differences in the flap area size or the choice of recipient vessels that would suggest an increased risk for elderly patients. The similarity in flap and recipient site parameters indicates that the SCIP flap procedure maintains its effectiveness and adaptability regardless of patient age.

The incorporation of diverse recipient sites and lack of functional rehabilitation data limit our ability to assess age-related recovery patterns. For example, in the case of limb reconstruction, elderly patients may require prolonged rehabilitation to restore mobility, but our retrospective design precluded such an analysis. Future prospective studies should integrate validated functional scales to address this gap.

Another consideration when using the SCIP flap in elderly patients is the potential for anastomotic thrombosis, given the smaller caliber of the SCIA and the increased prevalence of vascular sclerosis in this age group. Table 6 demonstrates that CTA abnormalities were more common in elderly patients, suggesting poorer vascular conditions in this population. However, there was no significant difference in the flap exploration rate or necrosis rate between the two groups. Our results indicate that although the SCIA has a relatively small caliber, it does not impact the success rate of SCIP flaps in elderly patients. One possible explanation is that atherosclerosis in elderly patients primarily affects larger blood vessels, while the smaller SCIA is typically less involved, ensuring better vascular patency after anastomosis. This may also explain why Professor Hong J. P. has recently advocated for perforator-to-perforator anastomosis in lower extremity and diabetic foot reconstructions [34,35].

The significance of vascular disease, particularly peripheral artery disease, in lower limb and head/neck reconstructions is well recognized, as compromised recipient vessels can critically impact surgical outcomes. In our SCIP flap cohort, preoperative CTA identified vascular abnormalities in most of elderly patients (*p* < 0.001), enabling tailored recipient vessel selection and intraoperative adjustments (e.g., proximal anastomosis to non-calcified segments). This strategy, combined with the SCIP flap’s anatomical adaptability, contributed to equivalent flap survival (*p* = 0.999) and reoperation rates (*p* = 0.678) across all groups, despite higher baseline vascular comorbidities in elderly patients. While our study included diverse indications, the limited representation of head/neck cases precludes definitive conclusions for this subgroup.

Perforator-to-perforator reconstructions of the lower limbs can be particularly valuable in overcoming certain limitations of the SCIP flap, such as the short pedicle, and in cases where recipient vessels have significant pathology, like peripheral artery disease, which is more common in elderly patients. A meta-analysis on the use of free flaps for perforator-to-perforator reconstructions of the lower limbs found that the SCIP flap was the most commonly used, accounting for over 56% of cases [36]. However, in our study, the use of perforator-to-perforator anastomoses was very limited. In most cases, we opted for anastomoses with the main recipient vessel or its primary branches. When the vessel calibers were similar, we performed end-to-end anastomoses, and when there was a significant diameter discrepancy, we generally used end-to-side anastomoses. Only when suitable perforator vessels were available at the recipient site did we consider perforator-to-perforator anastomoses, but such cases were infrequent in our cohort.

Elderly patients also exhibit diminished growth and healing capacities, making complications at the donor site a significant concern that warrants our attention. Our data showed that donor site complications were minimal and equally distributed between age groups, reinforcing the procedure’s safety profile. Two relatively serious complications of SCIP flap donor site are lymphatic leakage and lower limb lymphedema. However, we did not observe these complications in our study. The main technique to avoid these complications is to harvest the flap in the superficial fascia layer.

In our study, an AUC of 0.52 was found, indicating that age had a minimal and statistically insignificant impact on predicting flap survival outcomes. This method involved plotting the true positive rate against the false positive rate to determine the effectiveness of age as a predictor, with the resulting AUC reflecting the overall accuracy of the prediction. And this minimal predictive power highlights that the procedure’s safety and effectiveness are consistent across different age groups, including the elderly. The result underscores the success of flap reconstruction is not significantly influenced by age alone, reinforcing the procedure’s overall safety and reliability regardless of the patient’s age. Thus, the procedure can be confidently applied to elderly patients without major concerns about age-related differences in flap survival.

One limitation of this study is that the average age of the elderly patients was only 63.78 years, which is relatively young for this population. We lack data on patients aged over 80 or even 90, a subgroup that faces greater clinical challenges and uncertainties. Additionally, the broad categorization of baseline patient data may have overlooked significant nuances and variations in patient characteristics, potentially affecting the study’s findings and their applicability. Therefore, future research should aim to include a larger cohort of elderly patients to thoroughly assess the safety and efficacy of SCIP in this demographic.

Another limitation of our study is that the retrospective data collection constrained our ability to uniformly apply ASA scores across all cases. However, we analyzed surrogate markers of systemic risk, such as vascular disease and CTA abnormalities, both strongly associated with ASA III-IV status. Notably, 77.8% of elderly patients had abnormal CTA findings (*p* < 0.001), suggesting advanced physiological aging consistent with higher ASA grades. Even our study discussed age-related vascular comorbidities, future investigations should incorporate ASA scoring to better quantify physiological reserve and anesthesia risk. This would align with recent evidence showing that ASA classification outperforms age in predicting perioperative morbidity, particularly in populations with multimorbidity.

Overall, the data support the conclusion that the SCIP flap procedure is both safe and effective for elderly patients. Even though the absence of significant differences in key outcome measures, coupled with the ability to manage complex conditions associated with aging, underscores the procedure’s reliability and efficacy across different age groups.

## 5. Conclusions

This study shows the SCIP flap demonstrates consistent safety and effectiveness in elderly patients. Despite the higher prevalence of vascular disease in the Elder group, outcomes such as flap survival and postoperative recovery remain favorable, indicating that the procedure is a viable option for older patients. These findings support the continued use of SCIP flaps in reconstructive surgeries for aging populations, highlighting the need for tailored surgical strategies to optimize patient outcomes.

## Figures and Tables

**Figure 1 bioengineering-12-00394-f001:**
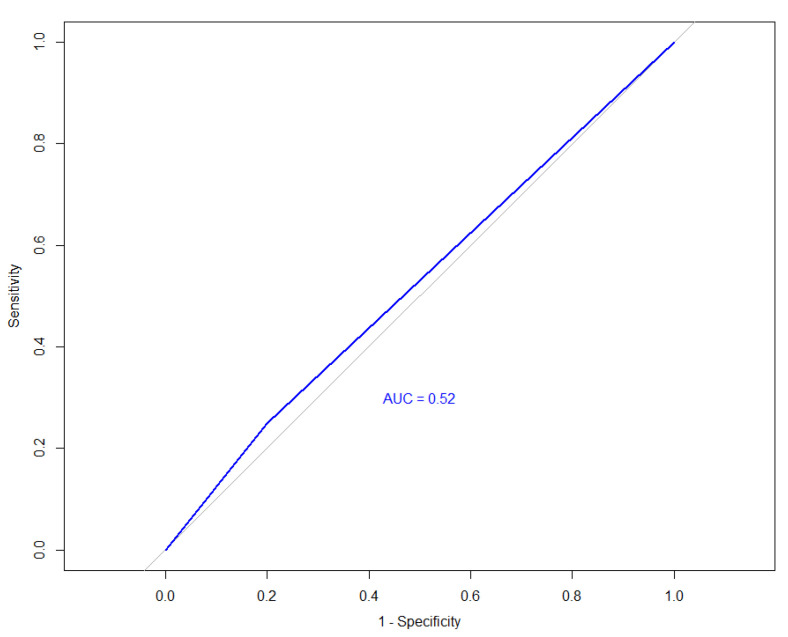
ROC curve for flap reconstruction results for young and elder patients. The blue curve represents the predictive curve of age on flap outcomes, while the gray line indicates the baseline for a random classifier.

**Table 1 bioengineering-12-00394-t001:** Baseline characteristics of the included patient’s cohort.

	Overall	Young (15–59 Years)	Elder (≥60 Years)	*p*-Values
Number (n)	37	28	9	
Gender = Male (%)	25 (67.6)	19 (67.9)	6 (66.7)	0.99
Age (year, mean (SD))	46.59 (15.27)	41.07 (13.31)	63.78 (3.49)	**<0.001**
Height (cm, mean (SD))	164.12 (9.17)	164.32 (9.73)	163.00 (6.00)	0.797
Weight (kg, mean (SD))	65.50 (15.09)	65.50 (16.97)	65.50 (7.33)	0.99
BMI (mean (SD))	26.41 (14.55)	26.30 (15.72)	26.97 (5.87)	0.934
BMI > 25 (%)	7 (18.9)	5 (17.9)	2 (22.2)	0.99

**Table 2 bioengineering-12-00394-t002:** Presentation of concomitant disease of the studied cohort of the patients.

	Overall	Young (15–59)	Elder (≥60 Years)	*p*-Values
Number (n)	37	28	9	
Tumor (%)	11 (29.7)	8 (28.6)	3 (33.3)	0.99
Vascular disease (%)	14 (37.8)	6 (21.4)	8 (88.9)	**0.001**
Lung disease (%)	1 (2.7)	0 (0.0)	1 (11.1)	0.544
Diabetes mellitus (%)	4 (10.8)	3 (10.7)	1 (11.1)	0.99
Infection (%)	15 (40.5)	10 (35.7)	5 (55.6)	0.506
Other concomitant diseases (%)	7 (18.9)	7 (25.0)	0 (0.0)	0.239

**Table 3 bioengineering-12-00394-t003:** CTA examination results and reasons for the surgery between two groups.

	Overall	Young (15–59)	Elder (≥60 Years)	*p*-Values
Number (n)	37	28	9	
Reasons for surgery (%)				0.247
Infections and complications	11 (30.6)	7 (25.9)	4 (44.4)	
Trauma and accidents	9 (25.0)	6 (22.2)	3 (33.3)	
Tumors	7 (19.4)	5 (18.5)	2 (22.2)	
Chronic conditions and other issues	9 (25.0)	9 (33.3)	0 (0.0)	
CTA results before surgery (%)				**0.001**
Abnormalities	10 (27.0)	3 (10.7)	7 (77.8)	
No abnormalities	14 (37.8)	13 (46.4)	1 (11.1)	
CTA not performed	9 (24.3)	8 (28.6)	1 (11.1)	
CTA not performed (ultrasound localized)	4 (10.8)	4 (14.3)	0 (0.0)	

**Table 4 bioengineering-12-00394-t004:** Summary of flap area, recipient sites, and recipient artery in reconstruction.

	Overall	Young (15–59)	Elder (≥60 Years)	*p*-Values
Number (n)	37	28	9	
Flap area (cm^2^) (mean (SD))	80.10 (55.51)	84.50 (60.58)	66.92 (35.98)	0.418
Recipient site (%)				0.297
Head/neck/trunk	9 (24.3)	6 (21.4)	3 (33.3)	
Lower extremities	22 (59.5)	16 (57.1)	6 (66.7)	
Upper extremities	6 (16.2)	6 (21.4)	0 (0.0)	
Recipient artery (%)				0.445
Ulnar artery	2 (5.4)	2 (7.1)	0 (0.0)	
Brachial artery perforator	1 (2.7)	1 (3.6)	0 (0.0)	
Posterior tibial artery	6 (16.2)	3 (10.7)	3 (33.3)	
Posterior tibial artery perforator	1 (2.7)	0 (0.0)	1 (11.1)	
Anterior tibial artery	3 (8.1)	3 (10.7)	0 (0.0)	
Intercostal perforator artery	1 (2.7)	1 (3.6)	0 (0.0)	
Superficial temporal artery	6 (16.2)	4 (14.3)	2 (22.2)	
Radial artery	1 (2.7)	1 (3.6)	0 (0.0)	
Radial artery nasolabial branch	1 (2.7)	1 (3.6)	0 (0.0)	
Medial knee perforator vessel	1 (2.7)	1 (3.6)	0 (0.0)	
Thoracoacromial artery	2 (5.4)	1 (3.6)	1 (11.1)	
Internal thoracic artery perforator	1 (2.7)	1 (3.6)	0 (0.0)	
Dorsalis pedis artery	9 (24.3)	8 (28.6)	1 (11.1)	
Medial plantar artery	1 (2.7)	1 (3.6)	0 (0.0)	
Lateral plantar artery	1 (2.7)	0 (0.0)	1 (11.1)	
Arterial end-to-end anastomosis (%)	22 (59.5)	15 (53.6)	7 (77.8)	0.37
Number of venous anastomoses (median [IQR])	2.00 [1.00, 2.00]	1.00 [1.00, 2.00]	2.00 [2.00,2.00]	0.181

**Table 5 bioengineering-12-00394-t005:** Revision and surgery outcomes of flap reconstruction.

	Overall	Young (15–59)	Elder (≥60 Years)	*p*-Values
Number (n)	37	28	9	
Length of hospital stay (mean (SD))	20.76 (5.97)	21.21 (5.95)	19.33 (6.16)	0.419
Flap Outcome =				0.999
Complete survival (%)	32 (86.5)	24 (85.7)	8 (88.9)	
Partial survival (%)	5(13.5)	4 (14.3)	1 (11.1)	
Total flap loss (%)	0 (0.0)	0 (0.0)	0 (0.0)	
Flap exploration (%)	8 (21.6)	7 (25.0)	1 (11.1)	0.678
Donor site complications (%)	1 (2.7)	1 (3.6)	0 (0.0)	0.999
Systemic complications (%)	1 (2.7)	0 (0.0)	1 (11.1)	0.544
Patient mortality (%)	0 (0.0)	0 (0.0)	0 (0.0)	-

**Table 6 bioengineering-12-00394-t006:** The two groups’ univariate analysis results shown for the odds ratios (OR) and confidence intervals (CIs).

	OR	95% CI	*p*-Value
Complete survival	1.333	0.164–28.127	0.809
Flap exploration	0.375	0.0185–2.621	0.392
Donor site complications	0.000	0.000–Inf	0.998
Systemic complications	2.328	0.000–Inf	0.997

## Data Availability

The datasets utilized and/or analyzed in this study cannot be made available due to organization policy and patient data privacy considerations.

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
