# Peer review of "Evaluation of Patient Benefits from the Superficial Circumflex Iliac Artery Perforator Flap in Elderly Patients"

_bioengineering, 2025, doi:10.3390/bioengineering12040394_

Round 1
Reviewer 1 Report
Comments and Suggestions for Authors
Some sentences are long and complex: please break down the longer sentences.
Some ideas are repeated in different ways.
Please correct minor grammar and syntax issues.
Introduction
The introduction is well-structured and relevant but can benefit from a modification.
Instead of just stating that no prior research exists, explicitly connect this gap to your study’s purpose.
Methods
The Materials and Methods section is well-structured.
Results
Some p-values are written inconsistently: it is recommended to follow a uniform format, such as italicizing p (p < 0.001).
When discussing the vascular disease prevalence difference (p = 0.001), it would be helpful to include an effect size (e.g., odds ratio or relative risk) .
Discussion
Comparisons with other free flap studies are mentioned, but more details would strengthen the discussion. For example were SCIP flap outcomes better, worse, or similar compared to other flaps in elderly patients?
The AUC value of 0.52 is briefly mentioned (lines 275–280), but its implication is unclear. It suggests only that age is a poor predictor of flap survival?
Conclusion
What specific strategies might be beneficial for elderly patients?
References Appropriate
Tables and figures good quality and appropriate
Comments on the Quality of English LanguageGood
Author Response
Please find the attached file for detailed response to the reviewer's comments.

Reviewer 2 Report
Comments and Suggestions for Authors
I commend the authors for this research. Extending indications for microsurgery in elderly patients is of crucial relevance as long as this can be achieved safely. However, I have some concerns which I hope can be addressed. Please find them listed below:
- My biggest concern is that the threshold of elderly individuals in this study is arbitrary and with the growing standards for quality of life and life expectancy, I have my reservations in considering 60 year-old “elderly”. You acknowledged this in your limitations. However, life expectancy in China is now close to 80 years. [PMID: 37004714] Conventionally, “elderly” has been defined as a chronological age of 65 years old or older, while those from 65 through 74 years old are referred to as “early elderly” and those over 75 years old as “late elderly.” I would suggest restricting your age distribution, by considering individuals aged 65 or older, and perhaps subdividing into two separate groups, as was done in previous studies on microsurgery in elderly patients (i.e. ref. 18). If not possible due to lack of data for patients older than 80 please acknowledge in more details in your discussion.
- Chronological age alone does not suggest much today as it used to in previous decades. In fact, preoperative evaluation now accounts for comorbidities, anesthesia risk (ASA score), “fitness for surgery”. It’s a pity that ASA score was not taken into account. Please discuss.
- Additionally, this study did not account for medical complications after surgery and length in hospitalization, which are also quite relevant in elderly and frail patients in general. Please acknowledge. Another limitation to the study is also how the aspect of functional rehabilitation after surgery could not be addressed due to the incorporation of various anatomical areas in the same study. i.e. How many warranted functional rehabilitation for reconstructions of the limbs? Did age affect duration or functional outcomes from rehabilitation?
- Was CTA performed preoperatively for the preoperative assessment of flap perforators, as what can be performed for other free flaps, [PMID: 36847143] or was it for to assess recipient vessels?
- The topic of microsurgery in the elderly has been discussed abundantly in literature, and your discussion should reflect this. Some of the earliest studies reported high rates of post-operative surgical complications, medical complications and even fatalities, despite selecting an arbitrary value of age above 50 to define “elderly”. [PMID: 1852818] This is in stark contrast with much more modern studies on very old patients which find either a correlation with medical complications but not surgical outcomes, [PMID: 16013064] or no significant difference at all regarding the rate of surgical or medical complications, flap failure and revision rate. [PMID: 27898199] Do you believe that mortality and complication rates have decreases and so significantly and microsurgery has become safer due to improvements in patient optimization, surgical improvements, reduction in surgical duration, anesthesia management? Please discuss.
- A study on 445 free flaps for breast reconstruction found that such procedures were safe in elderly patients as well. [PMID: 28061518] Several studies reported on the safety of free flaps for head and neck indications, where microsurgery has been particularly relevant in improving quality of life as well as functional outcomes. [PMID: 24323481; 25719702; 25820590; 26714952] One study reported on the safety in lower limb reconstructions, but did not address using the SCIP flap. [PMID: 33674054] I believe that this research could serve your discussion well. Consider implementing.
- Vascular disease and peripheral artery disease in particular are to be deemed relevant and significant especially in lower limb reconstructions where in some head and neck patients, where recipient vessels might be unusable and thus discarded. Because your study focused on a variety of indications for SCIP flaps, this should be addressed.
- In a meta-analysis on use of free flaps for perforator-to-perforator reconstructions of the lower limbs, the SCIP flap was found to be the most commonly used flap (>56% of cases). [PMID: 37394775] Perforator-to-perforator anastomoses can be used in specific conditions to overcome limitations of the SCIP flap such as the short pedicle, or limitations of the recipient vessels such as peripheral artery disease on larger caliber vessels, which are especially common in the elderly. Have any of the cases in your study required perforator-to-perforator anastomoses? Consider discussing.
Some minor errors which can be revised by requesting a native English speaker to proofread the manuscript.
Author Response

(The authors gave the same response as above.)

Round 2
Reviewer 2 Report
Comments and Suggestions for Authors
I am quite happy with the authors’ response to my comments. The transparency is appreciated, and as such your manuscript has been strengthened. Thank you.